# Multifocal Neuroblastoma and Central Hypoventilation in An Infant with Germline *ALK* F1174I Mutation

**DOI:** 10.3390/diagnostics12092260

**Published:** 2022-09-19

**Authors:** Anna Djos, Diana Treis, Susanne Fransson, Lena Gordon Murkes, Sandra Wessman, Jurate Ásmundsson, Agneta Markström, Per Kogner, Tommy Martinsson

**Affiliations:** 1Department of Laboratory Medicine, Institute of Biomedicine, Sahlgrenska Academy, University of Gothenburg, 405 30 Gothenburg, Sweden; 2Childhood Cancer Research Unit, Department of Women’s and Children’s Health, Karolinska Institute, and Pediatric Oncology, Astrid Lindgren Children’s Hospital, Karolinska University Hospital, 141 86 Stockholm, Sweden; 3Department of Pediatric Radiology, Astrid Lindgren Children’s Hospital, Karolinska University Hospital, 141 86 Stockholm, Sweden; 4Department of Clinical Pathology, Karolinska University Hospital, 141 86 Stockholm, Sweden; 5Department of Oncology-Pathology, Karolinska Institute, 171 77 Stockholm, Sweden; 6Pathology Department, Landspitali University Hospital, 101 Reykjavík, Iceland; 7Pediatric Neurology, Department of Women’s and Children’s Health, Karolinska Institute, 171 77 Stockholm, Sweden

**Keywords:** *ALK*, *PHOX2B*, neuroblastoma, neurocristopathies

## Abstract

A preterm infant with central hypoventilation was diagnosed with multifocal neuroblastoma. Congenital anomalies of the autonomic nervous system in association with neuroblastoma are commonly associated with germline mutations in *PHOX2B*. Further, the *ALK* gene is frequently mutated in both familial and sporadic neuroblastoma. Sanger sequencing of *ALK* and *PHOX2B,* SNP microarray of three tumor samples and whole genome sequencing of tumor and blood were performed. Genetic testing revealed a germline *ALK* F1174I mutation that was present in all tumor samples as well as in normal tissue samples from the patient. Neither of the patient’s parents presented the *ALK* variant. Array profiling of the three tumor samples showed that two of them had only numerical aberrations, whereas one sample displayed segmental alterations, including a gain at chromosome 2p, resulting in two copies of the *ALK*-mutated allele. Whole genome sequencing confirmed the presence of the *ALK* variant and did not detect any aberrations in the coding or promotor region of *PHOX2B.* This study is to our knowledge the first to report a *de novo*
*ALK* F1174I germline mutation. This may not only predispose to congenital multifocal neuroblastoma but may also contribute to the respiratory dysfunction seen in this patient.

## 1. Introduction

Neurocristopathies are a group of disorders that include tumors, tumor syndromes, malformations and other conditions caused by impaired development of neural crest cells (NCC). NCCs are highly migratory and multipotent progenitors that develop into a diverse range of cells, including neurons, glia, melanocytes and neuroendocrine cells, among others [1]. One neurocristopathy is neuroblastoma, a heterogeneous childhood cancer of the postganglionic sympathetic nervous system [2]. Neuroblastoma is mainly a copy number-driven neoplasm, where common genetic alterations include *MYCN* amplification, 17q-gain, 1p-deletion, 2p-gain and loss of 11q [3], with only a few genes showing recurrent genomic aberrations [4,5,6,7].

Most neuroblastoma tumors are sporadic, but a small subset of neuroblastoma cases are hereditary in an autosomal dominant manner, albeit with incomplete penetrance [8,9,10]. Patients with familial neuroblastoma often display multiple primary tumor sites [8,11,12,13]. In 2004, *PHOX2B* (paired-like homeobox 2b) was the first to be identified as a neuroblastoma predisposition gene [14,15], and in 2008, the *ALK* (anaplastic lymphoma kinase) gene was reported as the major cause of hereditary neuroblastoma [16,17], and shown to also be frequently mutated in sporadic neuroblastoma [16,17,18,19,20]. *PHOX2B* encodes a homeobox transcription factor that functions as a master regulator for neural crest differentiation into noradrenergic neurons, while ALK expression has been found in the developing central and peripheral nervous system, including sympathetic ganglia. One study indicated that PHOX2B directly regulates *ALK* transcription, thereby providing a direct link between these two neuroblastoma predisposition genes [21].

Germline mutations in *PHOX2B*, located at 4p13, do not only predispose to multifocal neuroblastoma, but are also often present in conjunction with other neurocristopathies, such as Hirschsprung disease (HSCR) and congenital central hypoventilation syndrome (CCHS), and are viewed as the main causative mutation of the latter [22,23,24]. Clinical manifestations have been shown to correlate to the type of *PHOX2B* mutation identified, such that expansion of the second polyalanine repeat is mainly associated with CCHS, while missense or frameshift mutations are associated with combined CCHS-HSCR-Neuroblastoma [25].

The *ALK* locus, spanning 728 kb at chromosome 2p23, encodes a membrane-associated receptor tyrosine kinase (RTK) that is highly expressed in fetal neuroblasts and has a role in early differentiation of neuronal tissues [26,27]. When deregulated, ALK also has high oncogenic potential, and aberrant ALK activation has been described in many different malignancies, including neuroblastoma [28]. The majority of reported *ALK* mutations in neuroblastoma are activating point mutations in the tyrosine kinase domain (TKD) that are present in 7–8% of all neuroblastoma cases, while amplification of the *ALK* locus occurs in approximately 2% of all sporadic neuroblastoma tumors. Recent analysis of relapsed neuroblastomas has shown an increased frequency of activating *ALK* mutations in these patients, and studies using deep sequencing revealed the occurrence of subclonal *ALK* mutations already at diagnosis, further pointing at the importance of ALK regulation in neuroblastoma etiology [29,30]. While a wide distribution of somatic *ALK* TKD mutations, including those affecting the three hot spot codons F1174, F1245 and R1275, have been observed in the *ALK* TKD, the most common germline mutation, R1275Q, has not been associated with any developmental anomalies. However, one study reported on two cases with severe encephalopathy and abnormal brainstem in co-occurrence with neuroblastoma associated with germline *ALK* F1174V and F1245V mutations, respectively, thereby indicating that these two specific mutations may cause serious disruption of the development of the central nervous system [31].

Here we report on a patient with multifocal, congenital neuroblastoma in association with clinical features similar to congenital central hypoventilation syndrome with a *de novo* germline *ALK* F1174I mutation in the absence of *PHOX2B* mutation.

## 2. Materials and Methods

### 2.1. Patient

The female infant was born preterm at gestational age 32 + 4 weeks and birth weight 1890 g by cesarean section due to imminent fetal asphyxia. After three minutes of mask ventilation, she started breathing spontaneously under continuous positive airway pressure (CPAP) support. A diaphragmatic hernia was suspected briefly after birth and the neonate was transferred to a tertiary center for surgical repair. At the neonatal intensive care unit, frequent episodes of apnea and desaturation were noted which did not respond to caffeine citrate and were beyond the expected degree of immature control of breathing. Defecations were also noted to be infrequent. 

A sparse movement pattern of the lower limbs along with muscular atrophy of the gluteic muscles prompted a magnetic resonance (MR) evaluation which revealed multiple lobular masses in the abdomen and thorax, suggestive of neuroblastoma (Figure 1A,B). Suspected intraspinal tumor growth could neither be confirmed nor safely ruled out. To reduce swelling and possible neural compression, it was decided to initiate chemotherapy treatment and intravenous steroid therapy without awaiting histopathology, and the child was started on etoposide and carboplatin combined with betamethasone. Efficient specific ALK-inhibiting therapy was not available at that time. Fine needle aspiration and core needle biopsy were performed two days later, showing a histopathology indicative of ganglioneuroblastoma (Figure 1C). Urinary catecholamine metabolites at diagnosis were partly elevated; dopamine/creatinine 1790 nmol/mmol (reference value > 2200), homovanillic acid (HVA)/creatinine 189 μmol/mmol (reference value < 20) and vanillylmandelic acid (VMA)/creatinine 300 (reference value > 11 μmol/mmol). The child developed clinical seizures which lacked corresponding electroencephalogram (EEG) changes, while interictal EEG showed epileptiform activity, and was also treated with midazolam and phenobarbital. 

Six weeks after the first course of etoposide and carboplatin, MR evaluation showed diminishing tumor volume. However, apneas and hypoventilation episodes continued without improvement, necessitating periods of (CPAP) and ventilator treatment. Hypercapnia was present at most times. Seizure-like episodes were also recurrent, though a cardiorespiratory polygraphy ruled out seizures as the cause of hypoventilation. At age three months, two months after diagnosis, with worsening episodes of desaturation and bradycardia likely induced by infection, cardiorespiratory resuscitation was unsuccessful, and the child died of respiratory failure. 

At autopsy, the partly neuroblastic, partly differentiated abdominal tumor mass still affected by anti-tumor therapy was observed to originate from the right adrenal gland. Immunohistochemistry showed tumor cells positive for CD56, NB84, INI-1, vimentin and partly positive for synaptophysin. Proliferation was 8–39%. Abdominal lymph node metastasis was detected, whereas left adrenal and pancreas were free from tumors. There was microscopic metastasis to liver portal zones. Of the thoracic manifestations, only the right tumor remained, showing a similar histology as the abdominal tumor (Figure 1D,F). Furthermore, central nervous system (CNS), skeleton, circulatory, urogenital and endocrine organs were found to be normal. Gastrointestinal organs were normal apart from the repaired diaphragmatic hernia. Small and large intestine displayed normal ganglia, ruling out HSCR (Figure 1G). Clinical characteristics are presented in Table 1.

### 2.2. ALK and PHOX2B Mutation Analysis 

Genomic DNA were extracted with DNeasy blood and tissue kit (Qiagen, Hilden, Germany) according to the manufacturer’s instructions. Mutation analyses of *ALK* TKD exons 21–25 and *PHOX2B* exon 1–3, with flanking introns, were performed with Sanger sequencing as earlier described [20,32]. All fragments were sequenced from both forward and reverse direction and analyzed using SeqScape v2.5 (Applied Biosystems, Waltham, MA, USA). Restriction enzyme digestion with ApoI (AATT) was used for confirmation of the presence or absence of the (3520T>A) base substitution. The reaction conditions were the following: 2 µL of PCR product was incubated with 0.75 U of ApoI enzyme (New England Biolabs, Ipswich, MA, USA) together with 1× BSA and 1× NEBuffer 3 at 50 °C for 3 h before products were visualized on a 2% agarose gel using UV-light and GelRed (Biotinum, Hayward, CA, USA).

### 2.3. High Resolution SNP Array Analysis

Genomic profiles of the three tumor samples were generated using Applied Biosystems CytoScan HD Array (Thermo Fisher Scientific, Waltham, MA, USA) according to the experimental procedure previously described [33]. The arrays detect both copy number and allele specific information at a high resolution. Whole genome copy number profiles were generated with the use of Chromosome Analysis Suite (ChAS) version 3.3 software (Thermo Fisher Scientific). [34].

### 2.4. Whole Genome Sequencing

Whole genome sequencing (WGS) was performed on Illumina xTen instrumentation at Clinical Genomics, SciLife Laboratories, Stockholm, Sweden, for an average coverage of 63X and 34X for tumor and constitutional DNA respectively, after TruSeq DNA PCR-Free library preparation (Illumina, San Diego, CA, USA) on DNA extracted from tumor sample 3 and corresponding constitutional blood. Read trimming, mapping to the human reference genome hg19 and variant calling were performed using the CLC Genomics Workbench 5.0 software (Qiagen, Aahus, Denmark). Only high quality called variants with a minimum 10% allele frequency for somatic calls and a minimum of 15% allele frequency for germline calls and a total read coverage of ten were considered for further analysis. Synonymous variants present in non-coding regions except those affecting canonical splice sites were discarded. Remaining variants were assessed manually through the Integrative Genomics Viewer (IGV) [35] for removal of calls due to mapping artifacts and paralogs. Copy number variant (CNV) identification was performed using the Canvas tool [36] with visualization of observed CNV calls in IGV.

## 3. Results

### 3.1. Analysis of ALK and PHOX2B

The co-occurrence of neuroblastoma and clinical suspicion of CCHS in this patient prompted us to sequence the coding parts of exon 1–3 and flanking splice sites of the *PHOX2B* gene. However, analysis of constitutional DNA extracted from blood showed no *PHOX2B* mutation (result not shown). Analysis of the ALK TKD (exon 21–25) for mutations was also performed. Sequencing of constitutional DNA from blood revealed an *ALK* missense mutation. The single base substitution T>A was located in exon 23 at base 3520 and it is predicted to result in an amino acid change from a phenylalanine to an isoleucine at position 1174 in the ALK protein (NM_004304.4; c.3520T>A; p.F1174I). Constitutional DNA samples (from blood) from the two parents were subsequently sequenced in order to determine whether the F1174I mutation was inherited or *de novo*. Sequencing showed that neither of the parents carried the mutation, i.e., both parents were homozygotes for the normal T allele. Tumor DNA from the patient was further analyzed showing that all three sampled tumor sites—sample 1, sample 2 and sample 3—contained the *ALK* F1174I mutation. A dose difference between the alleles could be seen in sample 3 which had an increased level of the mutated A-allele relative to the T-allele as compared to the other samples. In addition, a CNS DNA sample from the patient, obtained post mortem, was analyzed and it showed that the F1174I mutation was present also in the normal CNS tissue of the patient. Figure 2A shows the results of Sanger sequencing of tumor and normal material, respectively.

In order to verify that neither of the parents carried the mutation as well as to confirm the presence of the F1174I mutation in the blood lymphocytes of the patient, the PCR products were digested with the restriction enzyme ApoI. The heterozygous base substitution c.3520T>A found in the patient’s DNA is predicted to destroy the recognition sequence of ApoI (AATT) and hence, a mutated allele will not be cleaved while the normal allele will be cleaved. The results confirmed that none of the parents carry the c.3520T>A substitution since both their alleles were cleaved with ApoI (Figure 2B). The results also confirmed that the patient carried the mutation in a heterozygote form, i.e., both undigested and digested product were observed on the gel.

### 3.2. Copy Number Analysis with SNP Microarrays of Tumor Specimens

Results from array profiling of the three tumor samples showed that the samples had different genomic profiles. Two samples, sample 1 and sample 2, had a numerical-only profile with only whole chromosome gains and losses and no segmental aberrations. Although sample 1 and sample 2 both were numerical-only they showed a diverged pattern of chromosomal gains and losses (Figure 2C). In sample 1, chromosomes 3 and 12 had markedly more copies relative to the modal karyotype, while sample 2 had loss of chromosomes 4, 7, 14 and 21 relative to the modal karyotype. Sample 3 contained segmental aberrations, such as deletion of 1p, gain of chromosome 2p, a small deletion at chromosome 4q and a 13q-gain (Figure 2C). The 2p-gain seen in sample 3 is in concordance with the dose difference (2:1) seen on Sanger sequencing between the *ALK* mutated allele A and the reference allele T in this sample (Figure 2A,D). 

### 3.3. Whole Genome Sequencing on Constitutional and Tumor DNA

WGS confirmed the presence of a heterozygous *ALK* F1174I variant in constitutional DNA (Figure 3A; upper panel) and did not detect any aberrations in *PHOX2B*, including introns and promoter region. WGS was also negative for protein-changing alterations or segmental variants in germline DNA of other genes that have been implicated in neurocristopathies and/or in neural crest development (e.g., *NDN*, *ASCL1*, *MYO1H*, *LBX1*, *RET*, *GDNF*, *NRTN*, *SHH*, *NGF*, *SOX10*, *EDNRB* and *EDN3*). WGS of sample 3 showed consistency with the genomic profile from SNP-microarray with 1p-del, 2p-gain, 4q-del and 13q-gain, due to unbalanced translocations between chromosome 1 and 13 (chr1:g.pter_89862490delins[chr13:g.30926324_qter]) and chromosome 2 and 4 (chr4:g.175241605_qterdelins[chr2:g.pter_46688708]), respectively (Figure 3B). *ALK* variant allele frequency in sample 3 was 70% (Figure 3A; lower panel), consistent with a gain of the *ALK* mutated allele. Calling of somatic variants detected only two nonsynonymous alterations, one in *CEP170* (g. 243328110G>A, NM_014812.2: c.3152C>T p.(P953L)) with a coverage of 47X and variant allele frequency of 14.9% and one in *HIVEP2* (g.143093576G>T, NM_006734.3: c.2300C>A p.(P767H)) with a coverage of 67X and variant allele frequency of 20.9%.

## 4. Discussion

Neuroblastoma, CCHS and HSCR are all neurocristopathies caused by aberrations in growth, migration or differentiation of NCC. The term neurocristopathy was coined in 1974 by Robert Bolande who divided neurocristopathies based on clinical features, but recent reports propose a new classification that groups neurocristopathies according to the stage of neural crest development at the onset of disease [1]. The NCCs originate from four different segments of the anterior–posterior axis: cranial, cardiac, trunk and sacral segments. Both neuroblastoma and CCHS are considered trunk neural crest cell diseases caused by defects in neural trunk cell differentiation and/or migration, whereas HSCR is considered a disease originating in the cardiac and sacral neural crest cells and characterized by the failure of enteric neural crest cells to colonize the distal part of the intestine [1]. 

The patient, a newborn girl, had multifocal neuroblastoma of ganglioneuroblastoma histology as well as persistent apneas, hypoventilation and hypercapnia. This constellation of symptoms was reminiscent of CCHS, even if prematurity, preexisting diaphragmatic hernia with subsequent surgery requiring pain medication, the use of antiepileptic mediation and the generally poor medical condition posed a challenge to respiratory evaluation of the infant. Respiratory insufficiency did not improve in spite of diminished tumor volume in response to chemotherapy, indicating that hypoventilation was not caused by pulmonary compression. Rather, cardiorespiratory polygraphy findings, including the observation of disturbed heart rate control, imply a disruption of central autonomous regulation.

HSCR was clinically not evident and aganglionosis of the intestine was ruled out at autopsy (Figure 1G), although the patient did display delayed passing of meconium, distended abdomen and poor feeding. CCHS cases have also been shown to be associated with constipation in the absence of HSCR [37]. Likewise, the patient’s observed reduced heart rate variability, seizures, and possible deviant eyesight behavior have all been reported as conditions associated with CCHS [37,38]. 

The observed combination of symptoms led to suspicion of constitutional *PHOX2B* mutation. However, as sequencing of germline DNA detected no mutation in *PHOX2B*, sequential analysis of *ALK* led to detection of a heterozygous *de novo* germline F1174I *ALK* mutation. To our knowledge, this is the first reported case of a *de novo* germline *ALK* F1174I variant in association to neuroblastoma, and the co-occurrence with central hypoventilation merits special consideration. WGS of DNA from blood confirmed the lack of *PHOX2B* mutations and did not detect any other alterations in genes, such as *MYO1H* or *LBX1* that also could predispose to CCHS [39,40]. 

There are three hot-spot residues within the ALK kinase domain (F1174, F1245 and R1275) where most ALK mutations are found in neuroblastoma patients. The F1174I *ALK* variant has previously been shown to be a highly oncogenic gain-of-function mutation that causes activation of STAT3 and ERK in a ligand-independent manner at similar levels as F1174L [41]. Experimental data also show that *ALK* F1174 mutations have higher oncogenic capacity as compared to R1275 mutations [42] suggesting that respective alterations could come with differential clinical consequences also when emerging as germline mutations. Except predisposing to neuroblastoma, germline *ALK* mutations have rarely been associated with other developmental anomalies. However, one report describes two patients with multifocal neuroblastoma of neonatal onset, hypoventilation and severe nonepileptic encephalopathy with fatal outcome, that lacked mutations in the coding sequence of *PHOX2B* and instead displayed heterozygous *de novo ALK* mutations at F1245V and F1174V, respectively [31]. The authors observed an enlarged medulla oblongata in both these patients, which they suggest as indicative of germline *ALK* mutation in selected circumstances. However, the abnormal form of these children’s brainstems was not recapitulated in the patient presented here (Figure 1H). Still, these cases together with ours indicate that *ALK* germline mutations with higher oncogenic potential may contribute to a more severe clinical presentation with additional neurodevelopmental dysfunctions (see Appendix A for described similarities and differences between the three cases). The severity of *ALK* F1174L in neural development is supported by a study of a murine knock-in model where the mutation caused embryonic lethality when expressed prior to commitment to the sympathoadrenal lineage, while expression after lineage commitment instead impaired sympathetic progenitor differentiation and increased proliferation [43]. Lopez-Delisle et al. investigated the phenotype of knock-in mice bearing the Alk^F1178L^ mutation (corresponding to F1174L in human). In homozygous mice they observed high neonatal lethality, dramatically decreased milk intake, poor growth and slightly more apneas compared to littermate controls. Heterozygous knock-in mice did not show feeding or breathing difficulties. The authors concluded that ALK activation above a certain critical threshold is not compatible with survival in mice. Although unspecific, the described breathing and feeding difficulties seen in homozygote knock-in mice are in concordance with the observations made in the human patients with *de novo* germline activating *ALK* mutation. The knock-in Alk^F1178L^ mice did express *Alk* mRNA in the brainstem motor nuclei, but no abnormalities in size or shape of these structures were seen [44]. 

*PHOX2B* and *ALK* are both fundamental for the development of sympathetic ganglia, but very little is known about possible crosstalk between them. One study showed that PHOX2B binds to the *ALK* promotor and thereby participates in transcriptional regulation of *ALK* [21]. However, it is not known if the reciprocal could occur with ALK signaling affecting PHOX2B levels and thereby also cause perturbation of neuronal differentiation during specific developmental windows and give rise to neuroblastoma and a CCHS-like condition. As no additional alteration in genes associated with neuroblastoma or CCHS could be detected, we suggest that the *ALK* F1174I mutation may not only predispose to congenital multifocal neuroblastoma but could also have contributed to the respiratory dysfunction seen in this patient. Hence, further investigation of *ALK* genomic aberrations may be of interest in patients with a CCHS phenotype without *PHOX2B* mutations [38].

This patient had multifocal primary tumors localized left and right paravertebrally as well as in the abdomen (Figure 1A,B). At autopsy, material was harvested from the remaining thoracic tumor on the right side (sample 1), and from two distinct portions of the abdominal tumor (sample 2 and sample 3). SNP array profiling of the three tumor sites showed that two had only numerical aberrations, whereas one had segmental aberrations (Figure 2C). Sample 3 showed a gain at chromosome 2p (Figure 2C,D) that includes *MYCN*, *ALKAL2* and the mutated *ALK* allele as judged by both Sanger sequencing (Figure 2A) and by WGS (Figure 3A). WGS detected only two somatic nonsynonymous alterations affecting *CEP170* and *HIVEP2*, both genes without established connection to neuroblastoma or cancer in general. SNP-microarray analysis in combination with WGS indicates relatively quiet genomes with very low mutational burden, suggesting that oncogenic *ALK* may be sufficient for early neuroblastoma development.

## 5. Conclusions

This study reports the presence of a novel *de novo ALK* germline F1174I mutation in an infant presenting with multifocal neuroblastoma and central hypoventilation in the absence of *PHOX2B* alterations. Sequencing of *ALK* mutational hotspots should be performed in neonates presenting with neuroblastoma and hypoventilation without *PHOX2B* mutations.

## Figures and Tables

**Figure 1 diagnostics-12-02260-f001:**
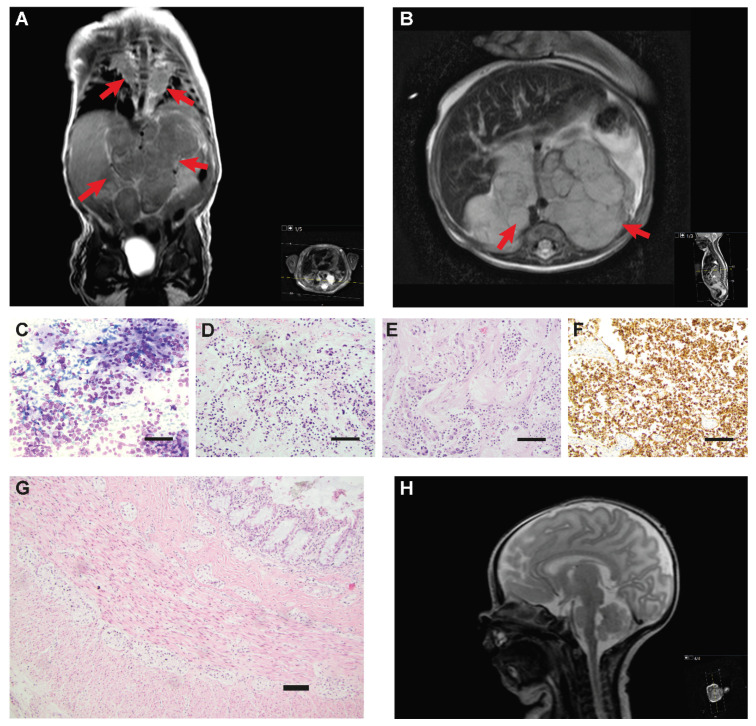
**Radiologic and pathohistological findings, brain MRI and colon histopathology.** (**A**,**B**) Thoracoabdominal magnetic resonance images (MRI), T1-weighted with gadolinium contrast in the coronal (**A**) and axial (**B**) plane, showing bilateral thoracic tumors as well as a large lobulated retroperitoneal mass (arrows). (**C**) Fine needle aspiration from abdominal tumor at diagnosis, May–Grünwald–Giemsa staining. Rosette formation and neuropil are seen; maturating ganglion cells were scarce. (**D**,**E**) Autopsy specimens from thoracic (**D**) and abdominal (**E**) tumor, hematoxylin-eosin staining. While the thoracic tumor shows an undifferentiated pattern with small round blue cells, the abdominal tumor displays areas with well-differentiated ganglion cells as well as areas with neuroblastic appearance. (**F**) Autopsy specimen from abdominal tumor staining positive for NB84. Scale bars, 100 μm. (**G**) Hematoxylin–eosin staining of autopsy specimen of the colon showing normal ganglia. Scale bar, 100 μm. (**H**) Sagittal T2-weighted MR image of the CNS showing normal conformation of pons and medulla oblongata.

**Figure 2 diagnostics-12-02260-f002:**
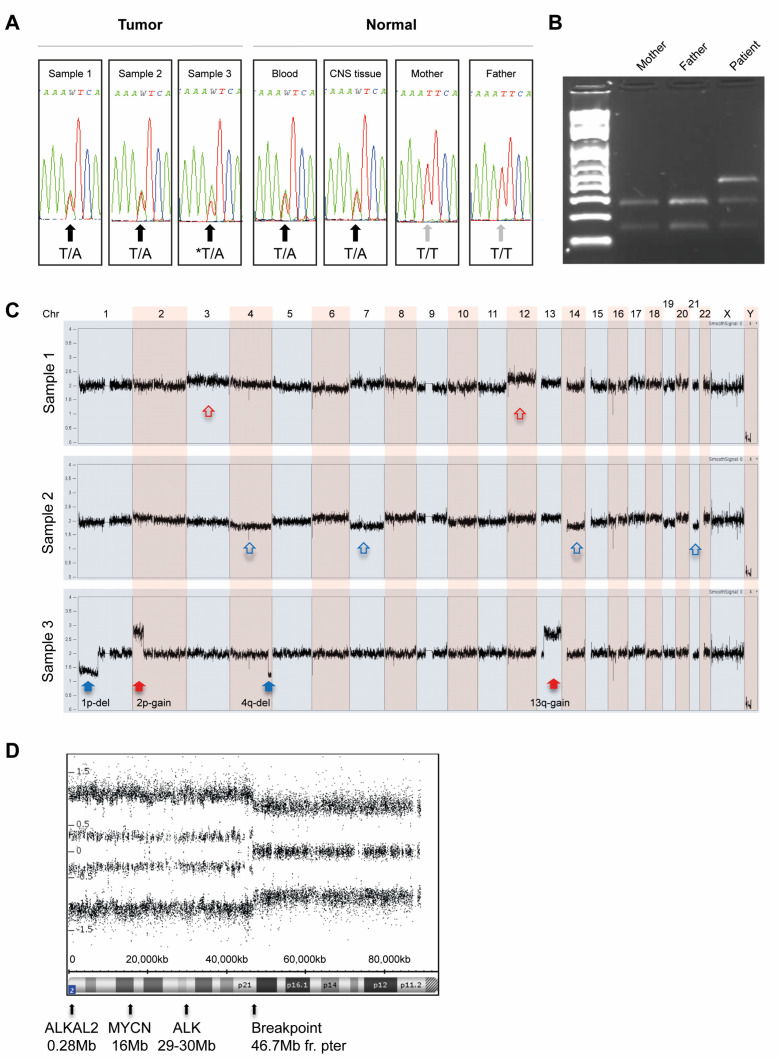
**Molecular analysis of the *ALK* gene and genomic profiling with SNP arrays.** (**A**) Electropherograms showing sequence covering the position corresponding to *ALK* F1174 in patient samples and parent samples. The heterozygote base substitution resulting in the *ALK* missense F1174I mutation is present in both tumor and normal tissue samples from the patient. Black arrows indicate samples with a heterozygote c.3520T>A base substitution and grey arrows point out samples lacking the mutation. The asterisk sign points out the sample where dose difference could be seen between the alleles. (**B**) Restriction enzyme digestion of the *ALK* exon 23 amplicon with ApoI verified the sequencing result. The heterozygous base substitution T>A will destroy the recognition sequence of ApoI (AATT); hence, a mutated allele will not be cleaved. Samples from left to right: blood from the mother, blood from the father and blood from the patient. (**C**) Genomic profiles of tumor DNA generated by SNP microarray. Genome-wide chromatograms of the three tumor samples are shown and aberrations are indicated with blue arrows for loss and red arrows for gain. The upper and middle profile show sample 1 and 2 that have numerical aberrations, i.e., only whole chromosome gains and losses. Although sample 1 and sample 2 were both numerical-only they were not identical. In sample 1 chromosomes 3 and 12 had markedly more copies (open red arrows) than average, while in sample 2 chromosomes 4, 7, 14 and 21 had less copies than average (open blue arrows). The bottom chromatogram shows the whole genome profile of sample 3. This tumor has segmental aberrations: 1p-deletion and 4q-deletion (filled blue arrows), 2p-gain and 13q-gain (filled red arrows). (**D**) Allele difference plot zoomed in on the chromosome 2p-arm of sample 3 showing the 2p-gain and the breakpoint at position 46,7Mb. Neuroblastoma-related genes included in the gained region are *ALKAL2*, *MYCN* and *ALK*.

**Figure 3 diagnostics-12-02260-f003:**
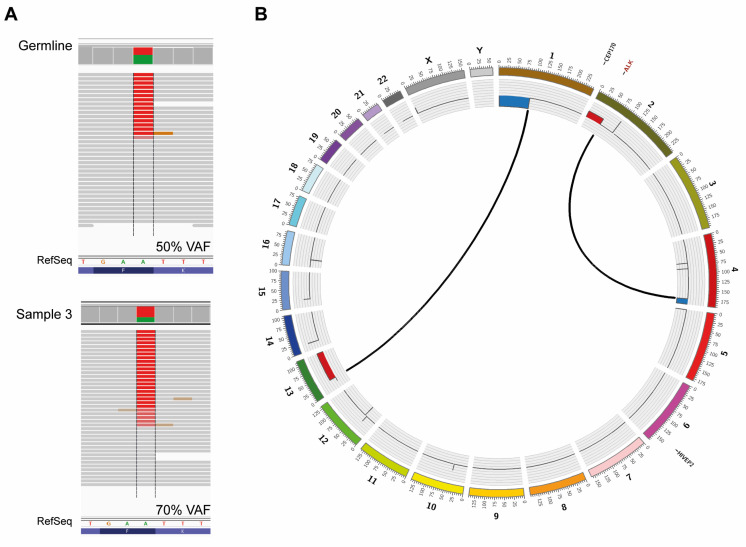
**Whole genome sequencing**. (**A**) WGS confirms presence of the *ALK* mutation in constitutional and tumor DNA of the patient. Read mapping of sample 3 and constitutional DNA are visualized in IGV. (**B**) Circos plot of sample 3 showing structural variants, copy number alterations, and somatic single nucleotide variants (SNVs). Copy number plots calculated based on the coverage ratio between tumor and corresponding normal tissue are shown on the inner circle, with gain of genomic material indicated in red and loss of genomic material indicated in blue. The lines within the inner circle indicate translocations between chromosomes, while genes affected by somatic SNVs are shown outside the outer circle in black, while the *ALK* gene, affected by a germline mutation, is shown in red.

**Table 1 diagnostics-12-02260-t001:** Clinical characteristics.

INRGSS Stage	Age at NB Diagnosis	INRG Risk Group	Histopathological Diagnosis	Genetic Profile	Ploidy
			Undifferentiated neuroblastoma in thorax	Sample 1: numerical only. No MNA or 11q-deletion.	4n
L2	4 weeks	Low	Ganglioneuroblastoma in abdomen	Sample 2: numerical only. No MNA or 11q-deletion.	4n
			Ganglioneuroblastoma in abdomen	Sample 3: other segmental aberrations. No MNA or 11q-deletion.	2n

## Data Availability

Not applicable.

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
