# Peer review of "Multifocal Neuroblastoma and Central Hypoventilation in An Infant with Germline ALK F1174I Mutation"

_diagnostics, 2022, doi:10.3390/diagnostics12092260_

Round 1

Reviewer 1 Report

In this manuscript, Djos et al. determined a de novo germline mutation in ALK gene (F1174I) in a NB patient with central hypoventilation. They reported that this is the first case with a de novo germline mutation of ALKF1174I in NB. They also concluded that this de novo mutation may contribute to the respiratory dysfunction seen in that patient.

This study is important in terms of understanding complexity of the genetical basis of neuroblastoma by adding a novel de novo mutation in ALK gene. The techniques, analysis methods, and also confirmative experiments are well-designed and adequate to come to such conclusion. Although the conclusion ‘this de novo mutation may contribute to the respiratory dysfunction seen in that patient’ seems to be specific to that patient, it may not be suitable to generalize this conclusion for all the NB cases in combination with central hypoventilation. Because they represent the results of another research group indicating a de novo ALKF1174I mutation in a NB patient without central hypoventilation. So it would be better to give a more accurate discussion of their findings in terms of their importance and possible contribution to the diagnosis, prognosis, and treatment of NB. In this way, they would emphasize the outcome of their study to be utilized in the future for NB. 

Reviewer 2 Report

Overall, this a well written study describing a case with ALK F1174I germline mutation that presented multifocal neuroblastoma and central hypoventilation.

Comments:

Authors should spell out abbreviations at first use. Authors should avoid use of abbreviation for a single word such as neuroblastoma "NB".

Clinical characteristics of the neuroblastoma tumor are not presented properly in the article. I suggest the authors to present the clinical characteristics such as MYCN status, ploidy, differentiation status, stage and risk in the form of a table. 

What other somatic mutations and structural variants were observed in the tumor sample?? List of all somatic alterations are required to be in supplementary table.

Did the patient show any improvement in respiratory dysfunction following reduction in tumor volume, if not is it probably due to gremlin alteration? I suggest authors to expand on this attribute in the discussion section.
